# Role of monkeys in the sylvatic cycle of chikungunya virus in Senegal

Benjamin M. Althouse[1,2,3], Mathilde Guerbois[4], Derek A.T. Cummings[5], Ousmane M. Diop[6], Ousmane Faye[6], Abdourahmane Faye[6], Diawo Diallo[6], Bakary Djilocalisse Sadio[6], Abdourahmane Sow[6], Oumar Faye[6], Amadou A. Sall[6], Mawlouth Diallo[6], Brenda Benefit[7], Evan Simons[7], Douglas M. Watts[8,9], Scott C. Weaver[10] & Kathryn A. Hanley[3]

Arboviruses spillover into humans either as a one-step jump from a reservoir host species into humans or as a two-step jump from the reservoir to an amplification host species and thence to humans. Little is known about arbovirus transmission dynamics in reservoir and amplification hosts. Here we elucidate the role of monkeys in the sylvatic, enzootic cycle of chikungunya virus (CHIKV) in the region around Kédougou, Senegal. Over 3 years, 737 monkeys were captured, aged using anthropometry and dentition, and tested for exposure to CHIKV by detection of neutralizing antibodies. Infant monkeys were positive for CHIKV even when the virus was not detected in a concurrent survey of mosquitoes and when population immunity was too high for monkeys alone to support continuous transmission. We conclude that monkeys in this region serve as amplification hosts of CHIKV. Additional efforts are needed to identify other hosts capable of supporting continuous circulation.

[1] Institute for Disease Modeling, Bellevue, 98005 WA, USA. [2] Information School, University of Washington, Seattle, 98105 WA, USA. [3] Department of Biology, New Mexico State University, Las Cruces, 88003 NM, USA. [4] Department of Microbiology and Immunology, University of Texas Medical Branch, Galveston, 77555 TX, USA. [5] Emerging Pathogens Institute, University of Florida, Gainesville, 32608 FL, USA. [6] Institut Pasteur de Dakar, Dakar, Senegal. [7] Department of Anthropology, New Mexico State University, Las Cruces, 88003 NM, USA. [8] Office of Research and Sponsored Projects, University of Texas at El Paso, El Paso, 79968 TX, USA. [9] Center for Biodefense and Emerging Infectious Diseases and Department of Pathology, University of Texas Medical Branch, Galveston, 77555 TX, USA. [10] Institute for Human Infections and Department of Microbiology and Immunology, University of Texas Medical Branch, Galveston, 77555 TX, USA. These authors contributed equally: Benjamin M. Althouse, Mathilde Guerbois. Correspondence and requests for materials should be addressed to B.M.A. (email: balthouse@idmod.org)

Arthropod-borne viruses circulating in enzootic cycles, i.e., cycles of alternating transmission between non-human reservoir hosts and arthropod vectors, pose the greatest risk of emergence into human populations of any class of pathogen. Here, modifying definitions from Haydon et al.[1], we define a reservoir host species as one in which a designated arbovirus is maintained permanently and which is required for the arbovirus to persist in nature. Multiple reservoir host species populations in a given area may exchange an arbovirus among them and thereby constitute a reservoir community. Spillover of enzootic arboviruses to humans may occur as a single-step transfer, mediated by a vector, from the reservoir host. Alternatively, an arbovirus may initially be transmitted from the reservoir host into a different amplification host species, one that supports robust replication of the virus but is not necessary for persistence of the virus, and then from the amplification host to humans[2,3]. For example, both West Nile virus (WNV) and Japanese encephalitis virus (JEV) are maintained in avian reservoir hosts, but WNV is frequently transmitted directly by mosquito vectors to humans from birds[4] while JEV is often amplified in pigs before causing human infections from vectors who feed on both pigs and humans[5].

Identifying the reservoir and amplification hosts of enzootic viruses is critical for predicting and, ideally, preventing human infections[6–8]. Moreover, there is burgeoning interest in the immune responses and genomics of reservoir hosts to zoonotic viruses[7,9]. However the host community for most enzootic arboviruses is incompletely characterized, and, in some cases, altogether unknown[10]. Moreover, with a few exceptions[11], transmission dynamics of most enzootic arboviruses within their enzootic hosts has not been quantified. This deficiency in knowledge of enzootic transmission cycles is perilous because of the tendency of some arboviruses, such as chikungunya virus (CHIKV), to emerge into human-endemic transmission cycles that can span the globe.

CHIKV circulates in two genetically distinct, enzootic, sylvatic transmission cycles in the forests of (1) West Africa and (2) East/Central/South Africa (ECSA)[12]. Sylvatic CHIKV periodically spills over into humans to cause individual cases and small outbreaks of disease in Africa. Some of these smaller outbreaks have proliferated into a human-endemic cycle in which transmission is enacted by the anthrophilic mosquitoes *Aedes aegypti aegypti* and *Aedes albopictus*. CHIKV causes significant morbidity in humans, including debilitating arthralgia and myalgia that can become chronic[13]. CHIKV epidemics in the Indian Ocean[14] and in India[13] have involved millions of cases; outbreaks on islands in the Indian Ocean, involving hundreds-of-thousands of cases and some fatalities, have included many tourists returning to Europe and the Americas with an estimated 2.4 million cases since 2013[13–16]. Recently, CHIKV established transmission in the Americas[15], and autochthonous CHIKV transmission has been documented in the continental United States[17].

Sylvatic CHIKV has been isolated from African green monkeys (*Chlorocebus sabaeus*), patas monkeys (*Erythrocebus patas*), Guinea baboons (*Papio papio*), guenons (*Cercopithecus aethiops*), and a bushbaby (*Galago senegalensis*) in Senegal[18–20]. Additionally sera from mandrills (*Mandrillus sphinx*) in Gabon[21], red-tail monkeys (*Cercopithecus ascanius schmidti*) in Uganda[22], and African green monkeys (*Cercopithecus (Chlorocebus) aethiops sensu lato*) and Chacma baboons (*Papio ursinus*) in South Africa and Zimbabwe[23] have tested positive for CHIKV antibodies. Together, these findings serve as the basis for the common

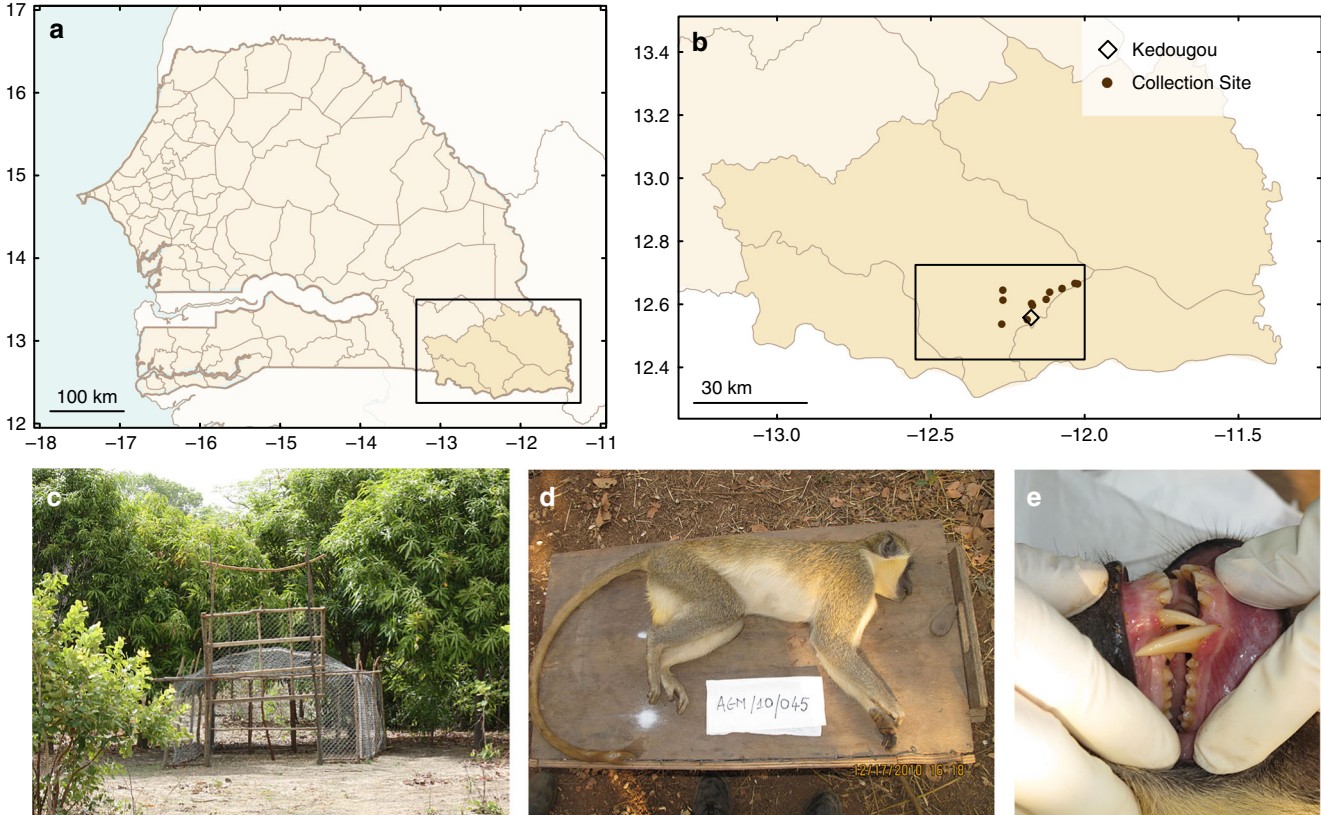

**Fig. 1** Monkey collection sites and sample individual. **a** A map of Senegal with the Kédougou Department boxed. **b** A map of the Kédougou region with the study region boxed and presented in detail in Fig. 2. **c** A typical trap, and **d**, **e** shows a male *C. sabaeus* estimated to be ~5 years (between 4 and 6 years) of age

assertion in the literature that non-human primates (NHPs) serve as the principal reservoir hosts of CHIKV[24]. However Chevillon et al.[20] have questioned this assumption, noting evidence of CHIKV infection in a wide variety of species in Africa, and Tsetsarkin et al.[25] have described NHPs as amplification hosts of CHIKV.

To elucidate transmission of CHIKV in its enzootic cycle, we leveraged data collected by the Institut Pasteur Senegal, which has conducted surveillance of sylvatic arboviruses and their mosquito vectors in the Department of Kédougou, Senegal (Fig. 1) over the past fifty years. Specifically, they have collected mosquitoes in sylvatic habitats of this region via human landing capture and screened them for arbovirus infection annually since the early 1970s. CHIKV has been isolated at roughly 4-year intervals over this timespan[26,27], primarily from mosquitoes in the genus Aedes (e.g., *Ae. furcifer*, *Ae. taylori*, *Ae. luteocephalus*, and *Ae. africanus*) [18]. We term mosquito species captured via human landing primatophilic, and periods when individual viruses are detected in primatophilic mosquitoes amplifications. During CHIKV amplifications, this virus has also been isolated from all three monkey species resident in Kédougou; African green monkeys, patas monkeys, and Guinea baboons.

We hypothesized that monkeys are the reservoir hosts for sylvatic CHIKV in Kédougou and that therefore the periodic amplification of sylvatic CHIKV detected in primatophilic mosquitoes is driven by depletion of susceptible NHP hosts during epizootics (epidemics in the reservoir hosts), local extinction of the virus, recruitment of susceptible hosts via births, and rein-troduction of the virus from NHP populations at distant sites[12]. To test this hypothesis, we conducted a 3-year study of the seroprevalence of CHIKV among individuals of known age from the three monkey species resident in the Department of Kédougou. These data were used to estimate key epidemiological parameters describing the transmission dynamics of CHIKV: age-specific seroprevalence, force of infection (FoI), and basic reproductive numbers in each of these three species. Contra our hypothesis, here we show that rates of CHIKV seropositivity in juvenile monkeys and CHIKV FoI were high in all three monkey species in periods between amplifications in primatophilic mosquitoes. These findings suggest that host species other than monkeys serve as reservoirs in this area, while monkeys instead act as amplification hosts. To our knowledge this is the first quantitative characterization of CHIKV transmission dynamics in its sylvatic cycle, the only age-stratified serosurvey of any arbovirus in NHPs, and the first time that this approach has been used to distinguish whether a particular species or group of species serves as reservoir host or amplification host for a zoonotic pathogen. Our findings will inform future work integrating data and models to assess risk to humans living near African sylvatic hotspots[28] as well as surveillance of potential enzootic CHIKV hosts outside of Africa.

## Results

**Monkey collections.** Monkeys were trapped at sites around Kédougou, Senegal (12°33 N, 12°11 W) close to the borders of Mali and Guinea (Fig. 1). Across all years of the study, 737 monkeys were collected in the 15 sites (Table 1). This included 219 *C. sabaeus*, 78 *E. patas*, and 440 *P. papio*. The PRNT results for the 117 NHPs collected in 2010 were previously published by Sow et al.[29], albeit solely to compare seropositivity among the three species and to note the temporal correspondence with human CHIKV infections in the region. Sites differed substantially in numbers of monkeys collected. *P. papio* were the most frequently collected species, but were only caught at 6 of the 15 sites (see Supplementary Table 1, Supplementary Figs. 1 and 2). *E. patas* were collected at 7 of 15 sites, and *C. sabaeus* at 9 of the 15 sites. Trapping sites were in close proximity to sites at

### Table 1 Monkeys collected by year

|  | 2010 | 2011 | 2012 | Total |
|---|---|---|---|---|
| *Chlorocebus sabaeus* | 52 | 78 | 89 | 219 |
| *Erythrocebus patas* | 34 | 4 | 40 | 78 |
| *Papio papio* | 103 | 200 | 137 | 440 |
| Total | 189 | 282 | 266 | 737 |

Table shows the numbers of monkeys collected per year across all sites

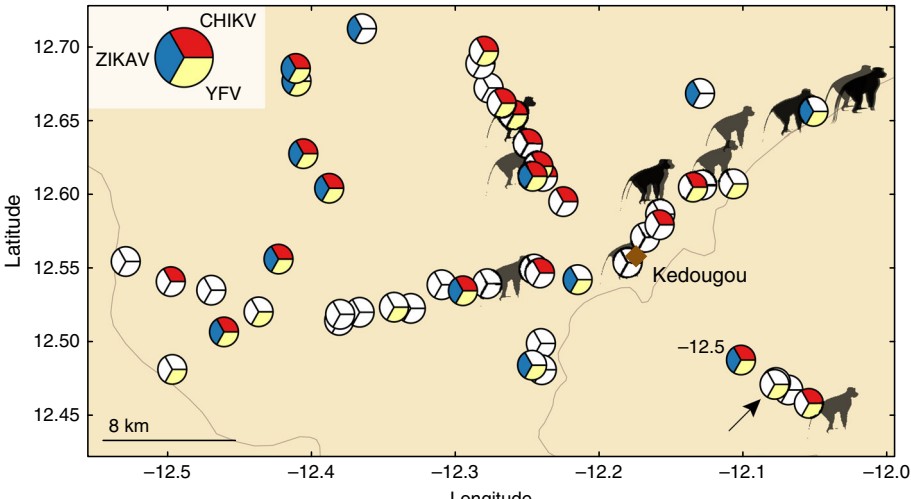

**Fig. 2** Distribution of monkey collection sites relative to chikungunya, yellow fever, and zika virus isolations from mosquitoes, 2009–2011. Figure shows the spatial distribution of monkey collection sites (monkey symbols) and the mosquito collection sites (pie charts). Pie slices indicate mosquito collection moving clockwise from 2009 at the top. Red indicates chikungunya virus (CHIKV) mosquito isolates in 2009, yellow indicates yellow fever virus (YFV) mosquito isolates in 2010, and blue indicates Zika virus (ZIKV) mosquito isolates in 2011; unfilled (white) slices indicate that there was no virus isolation in that year. Diamond indicates Kédougou town. Arrow indicates a ZIKV mosquito isolate that is obscured

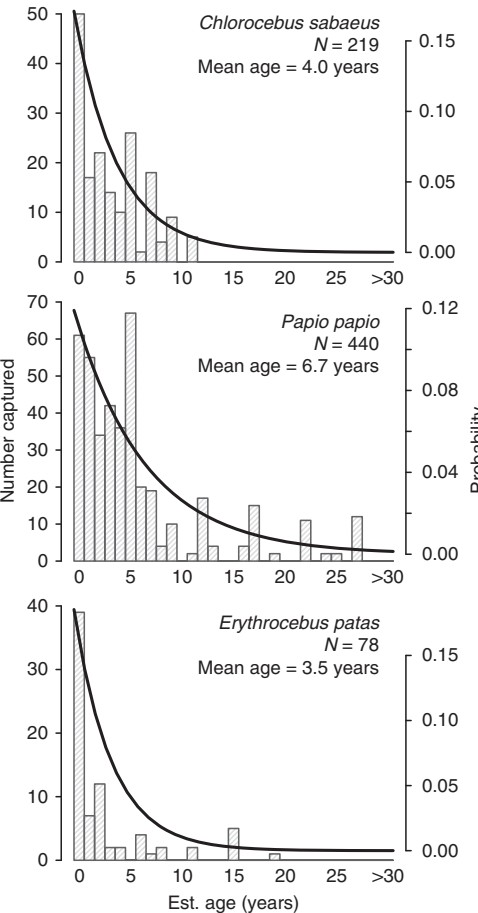

**Fig. 3** Age distributions of collected monkeys. Panels show the observed age distributions of collected monkeys with exponential distributions (thick line) with rates equal to the mean age of collected individuals, for *Chlorocebus sabaeus*, *Papio papio*, and *Erythrocebus patas*, respectively

| Table 2 Mixed effects logistic regression | |
|---|---|
| **Covariate** | **OR (95% CI)** |
| Intercept ($\beta_0$) | 0.26 (0.05, 1.24) |
| Age | 2.14 (1.84, 2.50) |
| *Erythrocebus patas* | 0.18 (0.05, 0.66) |
| *Papio papio* | 1.18 (0.45, 3.13) |
| Feb. collection | 1.51 (0.19, 11.78) |
| Mar. collection | 0.84 (0.17, 4.23) |
| Apr. collection | 1.34 (0.23, 7.81) |
| May collection | 1.86 (0.25, 13.87) |
| Dec. collection | 2.32 (0.28, 19.35) |
| Random intercept for Troop ($b_0$) | 0.72 (−0.69, 2.12) |
| ICC | 0.135 |

Table reports the estimates from a mixed effects logistic regression with CHIKV IgG seropositivity as the outcome and monkey age, species, month of collection as fixed effects and troop (same collection site and date) as a random intercept. Intercept corresponds to 0.26 probability of IgG positivity in the first year of life in *Chlorocebus sabaeus* primates collected in January, with the random effect indicating 95% of *Chlorocebus sabaeus* primate infants (<1-year-old) collected in January have PRNT$_{80}$ positivity rates between 0.060 and 0.52 ($\exp(\beta_0 \pm 1.96 \cdot b_0) / [1 + \exp(\beta_0 \pm 1.96 \cdot b_0)]$). ICC is the intraclass correlation for the random effect, and indicates about 13.5% of the total observed variance is due to variance within NHP troops

which mosquitoes were collected and screened for arboviruses in a concurrent study (Fig. 2)[30,31].

The mean ages of collected animals were relatively low, ranging from 3.5 years for *E. patas* to 6.7 years for *P. papio* (Fig. 3). These ages are consistent with previous estimates of the lifespan of wild *P. papio* and *E. patas*[32–34]. Ages were approximately exponentially distributed. As might be expected in collections biased toward juvenile animals[35], more male *C. sabaeus* (N = 147) and *P. papio* (N = 260) were collected than females (N = 70 *C. sabaeus* and N = 180 *P. papio* females), although more female *E. patas* (N = 64) were collected than males (N = 14). The sex of two individuals was not recorded. Captured females of all species were typically older than captured males (*C. sabaeus* 5.4 vs. 3.4 years [one-sided *t*-test, p = 0.0001], *P. papio* 8.7 vs. 5.4 years [p < 0.0001], and *E. patas* 3.9 vs. 1.6 years (p = 0.003).

**Seropositivity**. Rates of CHIKV seropositivity in all three species were high. Among 667 monkeys tested (198 *C. sabaeus*, 399 *P. papio*, and 70 *E. patas*) 479 (72%) were seropositive for CHIKV by PRNT. The remaining animals were not tested either because (i) adequate volumes of blood could not be drawn, (ii) identification data were not recorded, (iii) dental casts or photographs were inadequate for age estimation, or (iv) samples were lost during shipment. As expected during the dry season, no animals were positive for IgM antibody. Moreover, agreement between PRNT$_{50}$ and PRNT$_{80}$ was excellent, only 14 of 493 were positive

by PRNT$_{50}$ and not PRNT$_{80}$ at a cutoff of 1:20 (2% of all animals tested, Cohen's $\kappa = 0.95$ [95% CI 0.87–1]).

Mixed effects regression models were preferred to fixed effects models by AIC (432.7 vs. 446.0). Baseline seropositivity was high with the intercept and random effect indicating 95% of *C. sabaeus* primate infants (<1 year old) collected in January to have PRNT$_{80}$ positivity rates between 0.060 and 0.52 (Table 2). Age was strongly positively associated with odds of seropositivity (odds ratio [OR] = 2.14 [95% CI, 1.84–2.50] for each additional year of life), and *E. patas* had significantly smaller odds of seropositivity (OR 0.18 [95% CI, 0.05–0.66]) than the other two species. Intraclass correlation (ICC) calculated from the random intercept indicates about 13.5% of the total observed variance is due to variance within monkey troops.

**Antibody titer**. Antibody titers measured in PRNT are expressed as the maximum dilution that results in a given percent reduction in plaques, with a typical minimum cutoff of 1:20. Mixed effects linear regressions for the inverse PRNT$_{80}$ titers were preferred to fixed effects models by AIC (1670.3 vs. 1802.2) and are presented in Supplementary Table 8 and Supplementary Note 2. Age was significantly negatively associated with inverse titer, with each year of age corresponding to about a 4% decrease in titer ($\beta = 0.96$ [95% CI, 0.94–0.98]). This decrease is driven largely by *P. papio* in 2010 and 2011 (see Supplementary Fig. 3). Large differences in antibody titers were seen across study years, with 2011 having 75% lower titers ($\beta = 0.25$ [95% CI, 0.13–0.49]). This is likely due to there being no inverse titers of 1280 observed in 2011 (Fig. 4).

**O'nyong nyong virus seropositivity**. Forty-two randomly chosen monkeys (12 *C. sabaeus*, 25 *P. papio*, and 5 *E. patas*) were tested for O'nyong nyong virus (ONNV), and 16 (40%) had equivocal results (no consistent four-fold difference in reciprocal titers). The difficulty in distinguishing CHIKV from ONNV-induced immunity has been described previously[36]. Equivocal test results were not associated with age (OR: 0.92, 95% CI: 0.77, 1.09, p = 0.34), species (OR *E. patas*: 3, 95% CI: 0.4, 36.51, p = 0.3; OR *P. papio*: 0.42, 95% CI: 0.1, 1.73, p = 0.23), capture site and year (OR 2011: 0.8, 95% CI: 0.11, 5.2, p = 0.81; OR 2012: 0.26, 95% CI: 0.03, 1.77, p = 0.17), and dengue virus PRNT (OR: 0.81, 95% CI: 0.22, 2.94, p = 0.74). Only three monkeys with equivocal test results

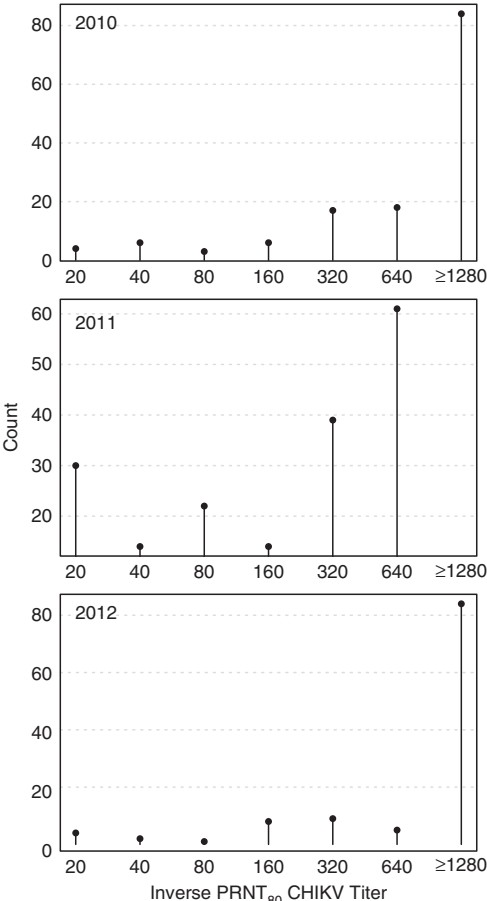

**Fig. 4** Inverse plaque reduction neutralization tests (PRNT$_{80}$) CHIKV titers by year. Figure shows number of animals (counts) per antibody titer by year for all three species of monkey

were under 2 years of age. We conclude from this analysis and from previous literature that the majority, and most likely all, of the positive PRNT values for CHIKV reflect CHIKV rather than ONNV infection.

**Force of infection**. In general, forces of infection (FoI; the rate at which susceptible individuals acquire infection) were high, ranging from 0.13/year (95% CI, 0.07–0.22) in *E. patas* in 2012, to well over 1 in *C. sabaeus* in 2011 ($\lambda(t) = 1.12$, [95% CI, 0.81–2.28]). Only two of the constant FoI models provided a better fit than the saturated model ($p > 0.05$; Chi-squared test comparing to the saturated likelihood, see Methods, Force of CHIKV Infection): *C. sabaeus* in 2010 and 2011. *P. papio* in 2010 was marginally better than the saturated model ($p = 0.05$; Chi-squared test comparing to the saturated likelihood, see Methods, Force of CHIKV Infection). As might be expected, the age-varying FoI was more flexible and provided a better fit (see Supplementary Table 8). FoI were high for younger monkeys, but there was a spike in FoI for monkeys aged about 8 years (Supplementary Fig. 5). Sensitivity analyses revealed the potential for overestimation of $\lambda(t)$ when the sampling is very biased by age (see Supplementary Figs. 8 and 9 and Supplementary Note 4).

Of particular note, there was high CHIKV seropositivity observed in young monkeys (≤2 years old) of all species in 2012—three years after the most recent CHIKV amplification detected in

mosquitoes (Figs. 2 and 5 and Supplementary Fig. 4 and Supplementary Note 3).

**Basic reproductive number**. Estimates of basic reproductive number ($R_0$)—the number of new monkey infections resulting from mosquito transmission from each infected monkey—varied by species, year, and assumed population structure. Assuming an exponential population structure with mortality rate equal to the inverse of observed mean ages, estimates of $R_0$ varied from 1.5 (95% CI, 1.3, 1.9) in *E. patas* in 2012, to 6.6 (95% CI, 5.1, 10.4) in *P. papio* in 2011. Generally, $R_0$ was highest in 2010 and in *P. papio*. *P. papio* consistently had the highest estimates of $R_0$ with estimates up to four times as high as either species in each year (see Table 3 and Supplementary Figs. 6 and 7).

## Discussion

In the Kédougou region, sylvatic CHIKV has been isolated from pools of primatophilic mosquitoes collected via human landing capture at roughly 4-year intervals since the early 1970s[18,26]. During these amplifications, outbreaks of CHIKV among humans occurred in Senegal in 1966, 1982, 1996, 2004, and in 2010[29,30,37], and the virus was isolated from humans in 1975 and 1983. We and others have hypothesized that monkeys are the reservoir hosts of CHIKV, and that during CHIKV amplifications, most susceptible monkeys are infected and rendered immune, so that the interval between CHIKV amplifications reflects the time needed for a sufficient number of susceptible monkeys to be born (susceptible recruitment)[26]. However, to date no studies have systematically examined the transmission dynamics of sylvatic CHIKV, or, to our knowledge, any arbovirus, in monkey hosts.

As expected based on its 4-year amplification cycle, CHIKV was isolated from 42 of 4211 mosquito pools collected across the Kédougou study region during the rainy season (June–January) of 2009. Infection rates among mosquito species differed temporally, with *Ae. furcifer*, *Ae. luteocephalus*, *Ae. taylori*, and *Ae. dalzieli* having significantly higher rates in December[37]. Despite similar mosquito collection efforts, and consistent with a 4.1-year periodicity in the CHIKV amplification cycle, the virus was not isolated from mosquitoes in the wet seasons of 2010, 2011, and 2012.

To assess whether susceptible NHP hosts were indeed depleted during this amplification, leading to local CHIKV extinction and consequent cessation of NHP infection, we initiated a 3-year age-stratified, serological survey of NHPs in Kédougou in 2010, immediately following the 2009 amplification. Over 700 NHPs were captured in the 2010, 2011, and 2012 dry seasons and we found high IgG seropositivity rates (72% by PRNT$_{80}$). Seroprevalence among monkeys in this study was dramatically higher than was reported in a recent study of CHIKV seroprevalence in East African non-human primates (13%)[38]. Catalytic models found correspondingly high forces of infection, in some cases approaching 1, making infection in the first year of life a near certainty. Even in 2012, 3 years after the last detected amplification of CHIKV in mosquitoes, we detected relatively high rates of infection in NHP infants (<1 year old), with seropositivity rates approaching 50% in those under 3 months old (see Supplemental Information). One interpretation of this finding is that infants are seropositive due to transfer of maternal antibody. However, while there is evidence of maternal transfer of CHIKV antibody in humans, the rates are not 100% and antibody levels decay rapidly[39,40]. Additionally, maternal transfer would be unlikely to sustain infant seroprevalence over several years. Thus we conclude that the majority of seropositive infants in this study were infected with CHIKV in their first year of life, despite the failure to detect infected primatophilic mosquitoes during these years. This finding contradicts the hypothesis that monkeys serve

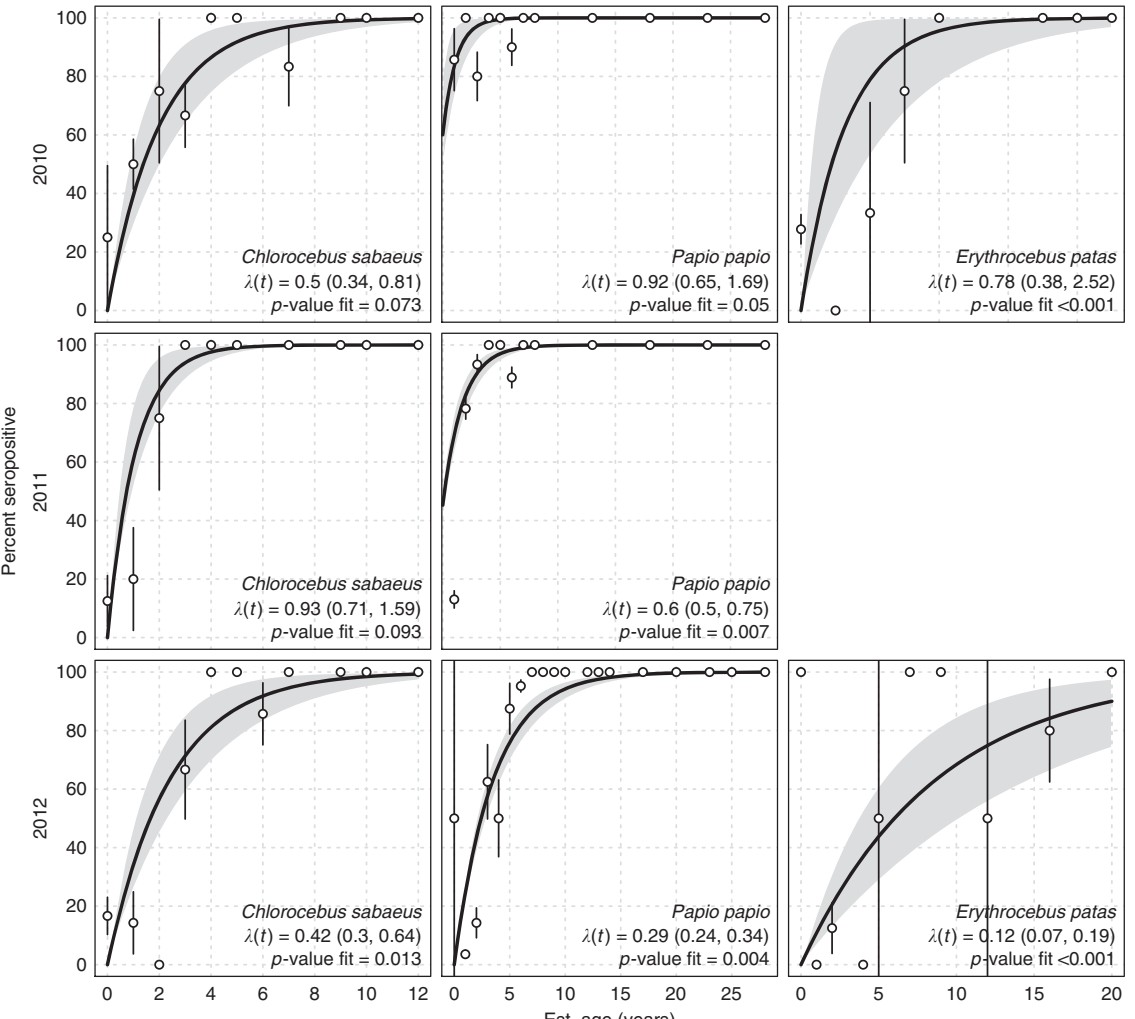

**Fig. 5** Forces of infection by species and year. Panels show the forces infection ($\lambda(t)$) and $p$-values (Chi-squared test comparing model to saturated model) for the fit across years for *C. sabaeus*, *E. patas*, and *P. papio*, respectively. Too few *E. patas* were collected in 2011 to obtain estimates. We included all monkeys <1-year-old in the 0-age category (we present seropositivity results for monkeys under 3 years of age in Supplementary Fig. 4). Points are the proportion of seropositive monkeys per age year with confidence intervals. Thick black line is the fit of the force of infection, gray bands are bootstrap confidence intervals for the fit

as reservoir hosts in the sylvatic CHIKV cycle in the Kédougou region and suggests instead that they act as CHIKV amplification hosts.

Our data suggest that an alternate cycle of CHIKV involving reservoir hosts other than monkeys and non-primatophilic vectors exists in Kédougou and is supporting CHIKV transmission. Although previous studies have suggested the existence of such cryptic reservoirs[20], our results provide the strongest evidence to date that the dynamics of CHIKV in monkeys preclude them from serving as reservoirs to maintain continuous CHIKV circulation in Kédougou. We do note that we have only investigated three species of NHP in this study; however, they are the most common NHPs in Senegal and the only three monkey species resident in the CHIKV-enzootic region we studied. CHIKV has been isolated from several small mammals in Senegal, including *Scotophilus* bats, a palm squirrel (*Xerus erythropus*), and a bushbaby (*Galago senegalensis*)[18,20]; moreover, bushbabies are important hosts of yellow fever virus in East Africa[41]. CHIKV may be maintained in cycles involving small mammals and non-primatophilic mosquitoes, which might not be readily detected by human landing capture methods. Indeed, Bosco-Lauth et al. found detectable CHIKV viremia in experimentally infected hamsters (*Mesocricetus auratus*), C57BL/6 mice (*Mus musculus*), and big brown bats (*Eptesicus fuscus*), indicating the possible roles of rodents and bats in CHIKV maintenance[42]. Hartwig et al. have also reported that amphibian and reptile hosts can sustain CHIKV viremia following experimental infection; of particular note is from this study is the Burmese python (*Python bivittatus*), an Old World species[43]. Alternatively, or in addition, it is possible that birds serve as a reservoir host for the virus. The source of bloodmeals from purportedly primatophilic mosquitoes that are known CHIKV vector species in Kédougou has been identified via PCR amplification of vertebrate cytochrome $b$[44]. This study found 60% (39 individual bloodmeals) of vector bloodmeals were taken from birds, with meals from Western Plantain-eater *Crinifer piscator* being the most common (26 bloodmeals, or 40% of the total). Primates accounted for 35% (23 bloodmeals) of the bloodmeals, and 5% (3 bloodmeals) of fed mosquitoes contained both human and Western Plantain-eater blood. Although previous studies discounted a possible role for birds as CHIKV hosts in India, where only the human-endemic cycle

**Table 3 Estimates of the basic reproductive number, $R_0$**

| Year | Age distribution | Chlorocebus sabaeus; $R_0$ (95% CI) | Papio papio; $R_0$ (95% CI) | Erythrocebus patas; $R_0$ (95% CI) |
|------|------------------|-------------------------------------|------------------------------|-------------------------------------|
| 2010 | Flat | 5.9 (4.1, 9.4) | 22.9 (16.8, 38.0) | 6.8 (3.5, 20.5) |
| | Literature Ages | 4.1 (3.0, 6.1) | 15.3 (11.5, 25.1) | 5.8 (3.1, 16.8) |
| | Mean Ages | 2.7 (2.2, 3.8) | 6.6 (5.1, 10.4) | 2.1 (1.6, 4.4) |
| 2011 | Flat | 10.7 (8.3, 17.5) | 15.7 (13.1, 19.0) | |
| | Literature Ages | 6.9 (5.5, 10.8) | 10.7 (9.1, 12.9) | |
| | Mean Ages | 4.3 (3.5, 6.4) | 4.0 (3.6, 4.7) | |
| 2012 | Flat | 5.0 (3.6, 7.5) | 7.8 (6.5, 9.3) | 2.5 (1.8, 3.8) |
| | Literature Ages | 3.5 (2.8, 5.0) | 5.6 (4.8, 6.6) | 2.2 (1.6, 3.0) |
| | Mean Ages | 2.4 (2.0, 3.1) | 2.5 (2.3, 2.8) | 1.5 (1.3, 1.9) |

Table reports the estimates of the basic reproduction number for the three species of monkey each year of the study period. Estimates are dependent on the assumed underlying population structure. "Flat" structure assumes a uniform population structure, "Literature Ages" and "Mean Ages" assume exponentially-distributed population structures with rates equal to the mean lifespan reported in the literature for captive monkeys, and the mean ages of the collected monkeys, respectively

is known to occur[45], further effort should be made to investigate the possible role of birds in the African enzootic cycle of CHIKV.

Making assumptions about the population structure of Senegalese NHPs, we determined the basic reproductive number of CHIKV in these populations to range from 1.6 to 6.6. Interestingly, we found large differences among species of NHP, with *P. papio* having estimates of $R_0$ up to three times that of the other NHPs. The forces of infection and reproductive numbers seen here indicate that all three of these species could initiate an explosive amplification of CHIKV in recently born monkeys who are susceptible to CHIKV. In geographic regions where sylvatic CHIKV transmission occurs, spillover into humans occurs frequently during CHIKV amplifications. Full emergence presumably is initiated when humans infected via spillover come into contact with the urban vectors *Ae. aegypti aegypti* and *Ae. albopictus*[46]. Thus, amplification hosts of CHIKV both directly and indirectly generate risk for human disease. In the last 60 years, CHIKV has emerged detectably into sustained human transmission only from the reservoirs in the ECSA sylvatic cycle[47], but the West African cycle has the potential to launch new CHIKV strains into urban transmission[25]. Maps of areas with high risk of spillover infection could be created if estimates of the range of movement and population numbers for the monkey species implicated as amplification hosts were known. Based on our estimates of force of infection, *P. papio* could be playing a larger role in the amplification of CHIKV in eastern Senegal than previously recognized, especially considering the substantial spatial heterogeneity mosquito density in the region[37]. In areas with low mosquito density, NHPs with higher forces of infection or values of $R_0$, may have a larger role in transmission[48].

Future studies should focus on identifying levels of CHIKV seroconversion and isolating CHIKV in species other than monkeys. Improved understanding of the enzootic, sylvatic cycle of CHIKV is essential to safeguarding the health of humans living in proximity to African foci of sylvatic transmission. Moreover the hunt for CHIKV reservoir hosts has increased in urgency since 2013, when CHIKV was introduced into the Americas. Recently, Lourenço-de-Oliveira and Failloux have shown that several neotropical, sylvatic, primatophilic mosquito species are highly competent vectors for CHIKV, opening the door to spill-back of CHIKV from humans to New World primates[49]. However it is possible, based on the data presented here, that other host species will also be required if CHIKV is to establish a sustained sylvatic cycle in the Americas[25,50].

## Methods

**Ethics statement.** All animal research was approved by the Institutional Animal Care and Use Committee (IACUC) of University of Texas Medical Branch, Galveston, protocol number: 0809063 (principal investigator: S.C.W.), and the entire protocol was approved on 27 November 2008 by the Consultative Committee for Ethics and Animal Experimentation of the Interstate School for Veterinary Sciences and Medicine, Dakar, Senegal (principal investigator: A.A.S.). No other specific permits were necessary. This approval is necessary and sufficient to conduct wildlife research in Senegal. Animals were trapped in large, open air containers (see Fig. 1c) with access to water and food, sedated and retained only long enough to take anthropomorphic measurements and draw a blood sample. Animals were released together as an intact troop upon recovery from ketamine anesthesia.

**Study site.** The Department of Kédougou comprises a mosaic of open savanna, woody savanna, outcrops of laterite (bowé), and relictual gallery forest, the latter concentrated along valleys and rivers. The Kédougou region is characterized by a tropical savanna climate, and receives an average of 1300 mm of total annual rainfall, with one rainy season from approximately June through November. Mean temperatures fluctuate around 25–33 °C throughout the year. Three monkey species reside in Kédougou: African green monkeys (AGM; *Chlorocebus sabaeus*), patas monkeys (*Erythrocebus patas*), and Guinea baboons (*Papio papio*). A relictual population of chimpanzees (*Pan troglodytes*) is present in the region[51], albeit in numbers too small to significantly affect CHIKV transmission. Senegal bushbabies (*Galago senegalensis*) are the only other NHP resident in Kédougou; populations sizes for this species in Senegal are not known[52]; because bushbabies are nocturnal and primarily consume arthropods it was not possible to collect them using the methods employed in this study. Humans in Kédougou have typically lived at low density (4/km²) in small dispersed villages. In the last 10 years, however, the region has experienced a "gold rush", and the expanding scope of mining operations is creating dramatic changes in population density, occupation and mobility[53].

The Kédougou area features a rich diversity of mosquito species including *Aedes aegypti formosus*, *Ae. africanus*, *Ae. centropunctatus*, *Ae. dalzieli*, *Ae. furcifer*, *Ae. hirsutus*, *Ae. luteocephalus*, *Ae. metallicus*, *Ae. neoafricanus*, *Ae. taylori*, *Ae. vittatus*, *Anopheles coustani*, *An. domicola*, *An. funestus*, *Culex poicilipes*, and *Mansonia uniformis*. *Ae. luteocephalus*, *Ae. taylori* and *Ae. africanus* show high rates of CHIKV infection but their distributions tend to be confined to forest canopies, thus they have been implicated in the maintenance of transmission of CHIKV among NHPs. *Ae. furcifer* has comparable CHIKV infection rates compared to the former three species, but a distribution that encompasses both the forest canopy and villages equally. We have therefore proposed that this species is the principal vector for spillover of sylvatic arboviruses into human communities around Kédougou[37].

**Monkey and mosquito collections.** *E. patas*, *C. sabaeus*, and *P. papio* were trapped during the dry season (generally December–May) in 2010, 2011, and 2012, from 15 sites in the Department of Kédougou (Fig. 1). Monkeys were captured in ground traps (see Fig. 1 and Supplemental Information) during the dry season, when other foods are scarce. Monkeys were sedated with 10 mg/kg of ketamine administered intramuscularly. Anthropological measurements were taken (weight, arm length, leg length, tail length, and body length), gender was determined, and nipple and scrotum conditions were noted. Dental casts and dental photographs were taken to assess which teeth were erupted (based on gingival emergence and complete eruption).

Monkey captures were conducted during the dry season, while mosquito collection was conducted during the rainy season (June–January)[37]. An amplification of CHIKV occurred in June 2009–January 2010, but CHIKV was not then detected in mosquitoes in 2010, 2011, or 2012. Yellow fever virus (YFV) and

Zika virus (ZIKV) were amplified in 2010 and 2011, respectively[30,31]. Figure 2 shows the relative location of the NHP sites and mosquito sites where CHIKV, YFV, and ZIKV were isolated.

**Determination of monkey age.** *Chlorocebus sabaeus*, *Erythrocebus patas*, and *Papio papio* were sorted into age classes based on the tooth eruption and degree of molar wear. The sequence of tooth eruption and molar occlusal wear was first determined separately for males and females of each species. Tooth presence, absence and gingival eruption information taken from casts and photographs were placed in order of tooth appearance to reveal the dental eruption sequence (see Supplemental Information). Published ages of dental eruption based on individuals of known age from captive and/or wild populations of the same species (*Chlorocebus aethiops* and *Erythrocebus patas*), or closely related species (*Papio cynocephalus* and *Papio anubis*) were used to estimate the chronological age of infant through young adult individuals in the Senegal populations[54–58]. See Supplementary Note 1, Supplementary Tables 2–7 for more information including age classes for NHPs used in this study.

**Serology.** Monkeys were bled from the inguinal vein while sedated and serum was frozen for later testing. Sera ware tested for CHIKV, dengue virus, and YFV antibody by plaque reduction neutralization tests (PRNT) to determine the dilutions of maximum sera that neutralized 50 and/or 80% of added virus[59]. PRNT80 data are presented here. O'nyong nyong virus (ONNV), an alphavirus with a close antigenic relationship to CHIKV, is present in Senegal. While antibodies raised against CHIKV will bind ONNV; antibodies raised against ONNV will not generally bind CHIKV[60]. This one-way antigenic cross-reactivity ensures the results presented here are likely true CHIKV antibody responses and not responses to ONNV[36]. However, a randomly chosen subset of samples were tested for ONNV by PRNT. We considered equivocal results if the ONNV antibody titer was greater than 4-fold larger than that for CHIKV. Penalized maximum-likelihood logistic regressions were run comparing equivocal to non-equivocal ONNV tests to look for biases based on NHP age, species, capture site and year, and dengue virus PRNT[61,62].

**Associations with CHIKV seropositivity.** To identify associations between NHP characteristics and CHIKV seropositivity, mixed-effects logistic and linear regressions were estimated. PRNT80 IgG seropositivity and inverse PRNT80 titers were the two outcomes of interest. Covariates of interest were NHP age, month of collection, and species, with NHP troop as a random effect to account for possible correlation of seropositivity at the troop level. As true NHP troops were not tracked, and indeed may not exist as consistent entities in some species, we considered those NHPs collected on the same day in the same site to belong to the same troop.

**Force of CHIKV infection.** Increases in seropositivity with age reflect the rate at which hosts acquire infection as a function of time as well as their risk of acquiring infection at different ages. The force of infection gives an indication of the intensity of transmission in a given area; high forces of infection indicating high prevalence of the pathogen in a population. Catalytic models of infection were fit to age-stratified data to determine annual forces of infection (denoted throughout as $\lambda(t)$). Models fit here are based on Grenfell et al.[63], and have been employed for dengue virus in Brazil[64] and Thailand[65]. Briefly, the proportion of the population susceptible to CHIKV infection of age $a$ at time $t$ is given by

$$x(a,t) = \exp\left(-\int_0^a \lambda(t-\tau)d\tau\right). \tag{1}$$

The proportion of individuals of age $a$ infected with CHIKV at time $t$ is

$$z(a,t) = 1 - \exp\left(-\int_0^a \lambda(t-\tau)d\tau\right) = 1 - x(a,t). \tag{2}$$

We can discretize the model by age and use maximum-likelihood methods for estimating $\lambda(t)$. The binomial log-likelihood (seropositive for CHIKV or not) of $\lambda_k(t)$ for age class $k \in [1, m]$ is

$$\ell(\lambda_k(t)) = \sum_{k=1}^m \left[n_{xk}\log[x(a_k,t_0)] + n_{yk}\log[1-x(a,t)]\right], \tag{3}$$

where $n_{xk}$ and $n_{yk}$ are the numbers susceptible and seropositive for CHIKV infection in age class $k$, respectively. We can compare the maximum likelihood estimates, $\ell_{max}$, to the saturated likelihood to estimate the goodness-of-fit of each model. The saturated likelihood, $\ell_{sat}$, is given by

$$\ell_{sat} = \sum_{k=1}^m \left[n_{xk}\log\left[\frac{n_{xk}}{N_k}\right] + n_{yk}\log\left[\frac{n_{yk}}{N_k}\right]\right]. \tag{4}$$

The statistic $X^2 = 2 \cdot (\ell_{max} - \ell_{sat})$ is $\chi^2$ distributed with $m - P$ degrees of freedom, where $m$ is the number of age classes and $P$ is the number of parameters

being estimated. As per Ferguson et al.[66], smaller $X^2$ values are better and models with $p$-values >0.05 are considered to fit the data well, as this indicates models that are statistically indistinguishable from saturated models. We calculated bootstrap confidence intervals to estimate uncertainty in estimates of $\lambda(t)$ by sampling NHPs with replacement and recalculating $\lambda(t)$. We estimate both constant and age-varying forces of infection.

**Calculating of the basic reproductive number of CHIKV.** The basic reproductive number, $R_0$, gives important information about the infectiousness of a pathogen in a population, and the feasibility of its eradication or control in that population. Higher values of $R_0$ would indicate higher numbers of infections and that a larger fraction of the population would need to be removed from the amplification pool (e.g., through vaccination or treatment) to stop transmission. $R_0$ can be calculated from $\lambda(t)$ if assumptions are made about the age structure of the population experiencing infection by using hazards to estimate the fraction of the population that remains susceptible and taking its reciprocal[66]. Let $f(a)$ be the fraction of the population aged $a$, and $w(a, t)$ be the fraction of the population aged $a$ exposed to CHIKV at time $t$, then

$$R_0 = \frac{1}{1 - \int_0^\infty f(a)w(a,t)da}. \tag{5}$$

We estimate $w(a, t)$ from $\lambda(t)$ as

$$w(a,t) = 1 - \exp\left(-\int_0^a \lambda(t-\tau)d\tau\right). \tag{6}$$

As the age structure of the NHP populations under study are not known, we assume three distributions of ages: Uniform(0, maximum observed age); Exponential(rate = 1/captive mean lifespan); and Exponential(rate = 1/mean observed age). We compared these to the observed age distributions of captured NHP. We use reported lifespans of NHP species in captive settings as an upper-bound on the lifespan.

**Sensitivity analyses.** Substantial sensitivity analyses were conducted to assess the effects of biased sampling by age; these are presented in the Supplementary Information.

**Data availability.** All relevant data are available from the authors.

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

## Acknowledgements

We are deeply indebted to Drs. Nicole Mideo (University of Toronto), Bill Messer (Oregon Health and Sciences University), and Rick Ostfeld (Cary Institute of Ecosystem Studies) for their improvements of the manuscript.

## Author contributions

D.A.T.C., D.D., M.D., A.A.S., D.M.W., S.C.W., and K.A.H. conceived the study. M.G., O. M.D., O.F., A.F., B.D.S., A.S., and O.F. captured primates and performed serology. B.B. and E.S. aged primates. K.A.H. compiled the data. B.M.A. and D.A.T.C. conceived the statistical analyses, B.M.A. performed these analyses. B.M.A. wrote the first draft of the manuscript. All authors approved and contributed to the subsequent drafts of the manuscript and agree with the results presented.

## Additional information

**Competing interests:** The authors declare no competing interests.

