## [Peer Review File · Nature Communications]

Reviewers' comments:

Reviewer #1 (Remarks to the Author):

This paper is well written with few errors which are listed below. The authors studied the role of CHIKV transmission of three monkeys over a three year period in Senegal. Although this is an interesting study, the overall interest to a wider community is questionable. One item of note was how samples were lost during shipment on line 100.

Editorial comments:

Overall comments - use more primary references
check for italicized errors

Please correct the following:

line 65 "Contra our"

Figure 2. ZIKAV

line 123 ONNV - is this the first use?

Reviewer #2 (Remarks to the Author):

General comments:

This manuscript represents an important addition to the sparse data concerning CHIKV sylvatic transmission and maintenance. It reports on the age-distribution of seropositivity, using newly seropositive infant monkeys as a means of estimating "recent" transmission intensity to address the hypothesis that CHIKV is maintained in this sylvatic cycle. While generally well written and methodologically appropriate, there are some minor concerns.

Specific comments:

In line 105, the statement "Baseline seropositivity was high with the intercept and random effect indicating..." First, this is an unnecessarily overcomplicated way to indicate a 95% confidence interval (which is based on much more than general variability, akin to a standard deviation). Second, random effects do not indicate anything as they are just a measure of general variability. Do you mean random intercept? Please clarify and changes should also be made to the table legend as the data is included in the legend but not in the Table. Either include in the table or remove from the legend. Also, the intercept is the probability of IgG positivity when all X's are held constant at 0. Thus, singling out the single variable "age" seems not well justified.

40% is not an insignificant proportion of monkeys to cross-react with ONNV. It does not seem that this would lend quite as much confidence in the specificity of PRNT for CHIKV as the authors state. Consider softening this assertion and discussing further in Discussion.

For the force of infection modeling – was there an undertaking of the age-varying model stratified over monkey species to determine whether a specific species was more likely to act as a reservoir rather than the entirety of the NHP biodiversity?

First sentence: Enzootic cycles don't pose greatest risk of any other pathogen (that's how the

sentence reads) – consider: “Vector-borne viruses with enzootic cycles...”

Line 5: should refer to “reservoir host species population” if you’re talking about permanent maintenance so as not to confuse with persistent viremia, etc.

Line 13-14: should include the mosquito in the statement re: JEV

There are several cases of italicized words (line 2-3, lines 9-10) – is this a journal keyword specification? Else, italics should be removed when not necessary.

Defining all NHP as amplification hosts rather than reservoirs is premature given that :

1. there was no apparent effort made to isolate virus from serum – thus all monkeys are presumed negative, even seronegative individuals
 2. is based on mosquito trapping data to define periods of amplification (a process mired in uncertainty)
 3. admittedly limited to three monkey species in the area studied
- Thus, the authors have data to indicate these species are likely not reservoirs in this area, but cannot extrapolate to the entirety of NHPs.

Reviewer #3 (Remarks to the Author):

Review: Althouse et al. Nature Communications

Title: Role of monkeys in the sylvatic cycle of CHIKV in Senegal

Take home: Even very young monkeys are all seropositive. This means that they are probably all exposed to infection, and seroconvert at an early age. This renders them immune to infection starting very early, and they therefore do not pass infection on to mosquitoes.

My main confusion with the findings of the paper I think stem from the confusion between amplification host and reservoir host. I looked up a few other papers to try to get a sense for how this term is used elsewhere. At least two references I found (Weaver 2005, <https://www.ncbi.nlm.nih.gov/pubmed/16358422>; Vorou 2016 <http://www.sciencedirect.com/science/article/pii/S1201971216310578>) clarified the definition, but both of these differed (I think) from how Althouse et al. seem to be using the term. Does an amplifying host amplify viremia and therefore increase the likelihood that vectors become infected (the definition put forward by these two references)? Or does amplification refer to high levels of virus neutralization, such that amplification hosts both indicate that there is plenty of virus around that needs neutralizing, but in mounting this response they prevent onward transmission to vectors? It is clear that the authors are purporting the latter in their paper (abstract). But how is it an amplification host if it’s not actually passing on the virus?

In line 10, authors clearly define amplification hosts as those that support robust viral replication, and state that arbovirus transmission into humans is facilitated by these amplification hosts (this definition squares with the other two references). But in the abstract (third sentence from the end), authors state that because immunity (seropositivity) is too high, they must be amplification hosts because they do not support transmission. How can they be amplification host for humans if they are all immune, and therefore not passing CHIKV on to vectors? These statements seem in direct opposition to one another. If anything, the primates sampled in this study are very sensitive indicators that CHIKV is present in the environment – indeed, they seem more sensitive an indicator than primatophilic mosquitoes. But the NHPs are not playing a significant role in the transmission cycle of CHIKV in this system. Again, in line 68, authors state that high CHIKV seropositivity between amplifications in mosquitoes suggests that “host species other than

monkeys serve as reservoirs in this area, while monkeys instead act as amplification hosts" – I agree with the first part – this evidence certainly suggests these species are not the reservoirs for CHIKV in this system. But the assertion that they must instead be amplifying virus seems counter to the previous assertion that these species are all immune and therefore not transmitting CHIKV to either vectors or to humans. This confusion followed me throughout the paper, and I had a hard time squaring lines 215-232 – if the majority of the populations are becoming infected and maintaining immunity starting at an early age (within the first year of life), how are these species at risk of initiating an explosive amplification of CHIKV?

I thought the field work and modeling were quite rigorous and clearly presented. I suspect that my lingering confusion over the inference from otherwise very straightforward empirical results could be easily remedied with clarifications in key places in the text. Perhaps a new definition of "amplification host" could specify how the immune response (neutralizing persistent CHIKV, irrespective of amplifications detectable in mosquitoes) can be decoupled from transmission probability (the likelihood of serving as competent hosts from mosquito vectors). It seems to me like these two components of the definition are difficult to consider independently of one another because doing so would then lead to very different ecological outcomes.

We wish to thank the reviewers for their thoughtful comments on our work. Please find *the reviewer's comments in italics*, and our responses in normal face.

=====

Reviewer #1 (Remarks to the Author):

This paper is well written with few errors which are listed below. The authors studied the role of CHIKV transmission of three monkeys over a three year period in Senegal.

Although this is an interesting study, the overall interest to a wider community is questionable.

With respect, we believe that microbiologists, immunologists, medical entomologists, ecologists and evolutionary biologists will be deeply interested in this study, as will any other scientist with an interest in emerging viruses.

One item of note was how samples were lost during shipment on line 100.

Samples had to be shipped from the site of blood collection in the field to the field lab in Kédougou, Senegal and from the field lab in Kédougou to Dakar, and from Dakar to the US. Some tubes with samples were damaged or misplaced at each of these steps, but there was no bias evident in the loss of the samples.

Editorial comments:

Overall comments - use more primary references

We have tried to use primary references as often as possible. See additionally response to reviewer 3.

check for italicized errors

We are not exactly sure to what the reviewer is referring.

Please correct the following:

line 65 "Contra our"

Figure 2. ZIKAV

line 123 ONNV - is this the first use?

We have fixed these.

Reviewer #2 (Remarks to the Author):

General comments:

This manuscript represents an important addition to the sparse data concerning CHIKV sylvatic transmission and maintenance. It reports on the age-distribution of seropositivity, using newly seropositive infant monkeys as a means of estimating “recent” transmission intensity to address the hypothesis that CHIKV is maintained in this sylvatic cycle. While generally well written and methodologically appropriate, there are some minor concerns.

Specific comments:

In line 105, the statement “Baseline seropositivity was high with the intercept and random effect indicating...” First, this is an unnecessarily overcomplicated way to indicate a 95% confidence interval (which is based on much more than general variability, akin to a standard deviation). ...

We introduce the random intercept model to adjust for potential between-troop variability. The reported 95% confidence interval was calculated and reported specifically to give the reader an estimate of overall seropositivity adjusting for potential confounding factors.

... Second, random effects do not indicate anything as they are just a measure of general variability. Do you mean random intercept? Please clarify and changes should also be made to the table legend as the data is included in the legend but not in the Table. Either include in the table or remove from the legend. ...

Yes, we did mean random intercept. We have modified the table accordingly.

... Also, the intercept is the probability of IgG positivity when all X’s are held constant at 0. Thus, singling out the single variable “age” seems not well justified.

The intercept is an estimate of the rate of seropositivity for *Chlorocebus sabaenus* primates of age <1 collected in January, adjusting for potential between-troop variability. We highlight the variable “age” as the odds ratio for seropositivity for each additional 1 year of life, adjusted for monkey age, species, month of collection, and potential between-troop variability. We hope this clarifies the estimates for the reviewer.

40% is not an insignificant proportion of monkeys to cross-react with ONNV. It does not seem that this would lend quite as much confidence in the specificity of PRNT for CHIKV as the authors state. Consider softening this assertion and discussing further in Discussion.

While yes, 40% may not be considered insignificant, we find equivocal testing results to not be statistically significantly associated with monkey age, species, capture site and year, and corresponding dengue virus PRNT. Moreover, O'nyong nyong virus antigen can elicit a CHIKV antibody response but CHIKV does not elicit a strong O'nyong nyong virus antibody response (Blackburn et al, Antigenic relationship between chikungunya virus strains and O'nyong nyong virus using monoclonal antibodies, Research in virology, 1995). Together these two factors strongly suggest that the CHIKV antibody responses detected in this study

reflect solely CHIKV rather than mixed CHIKV and O'nyong nyong infections (Powers et al, Re-emergence of chikungunya and o'nyong-nyong viruses: evidence for distinct geographical lineages and distant evolutionary relationships, Journal of General Virology, 2000).

For the force of infection modeling – was there an undertaking of the age-varying model stratified over monkey species to determine whether a specific species was more likely to act as a reservoir rather than the entirety of the NHP biodiversity?

Yes, we refer the reviewer to the supplemental information, Figure S5. We find high forces of infection in young monkeys and in monkeys around 8 years of age.

First sentence: Enzootic cycles don't pose greatest risk of any other pathogen (that's how the sentence reads) – consider: "Vector-borne viruses with enzootic cycles..."

We agree the sentence was confusing. We have modified it slightly:

"Arthropod-borne viruses in enzootic cycles, i.e. cycles of alternating transmission between non-human *reservoir hosts* and arthropod vectors, pose the greatest risk of emergence into human populations of any class of pathogen."

Line 5: should refer to "reservoir host species population" if you're talking about permanent maintenance so as not to confuse with persistent viremia, etc.

We have modified the sentence accordingly.

Line 13-14: should include the mosquito in the statement re: JEV

We have included the mosquito in the statement.

There are several cases of italicized words (line 2-3, lines 9-10) – is this a journal keyword specification? Else, italics should be removed when not necessary.

We wish to highlight the terms *reservoir host* and *amplification host* as we define them to stress to the reader their importance for the manuscript. We believe that the confusion expressed by reviewer 3 (which is entirely justified given the lack of a consistent definition for these terms in the field at large) emphasizes the need for these italics.

Defining all NHP as amplification hosts rather than reservoirs is premature given that :

- 1. there was no apparent effort made to isolate virus from serum – thus all monkeys are presumed negative, even seronegative individuals*
- 2. is based on mosquito trapping data to define periods of amplification (a process mired in uncertainty)*
- 3. admittedly limited to three monkey species in the area studied*

Thus, the authors have data to indicate these species are likely not reservoirs in this area, but cannot extrapolate to the entirety of NHPs.

We take the reviewers point, and have modified the abstract and discussion to modify our conclusion to state that monkeys in the Kedougou region do not serve as reservoir hosts.

Reviewer #3 (Remarks to the Author):

Review: Althouse et al. Nature Communications

Title: Role of monkeys in the sylvatic cycle of CHIKV in Senegal

Take home: Even very young monkeys are all seropositive. This means that they are probably all exposed to infection, and seroconvert at an early age. This renders them immune to infection starting very early, and they therefore do not pass infection on to mosquitoes.

My main confusion with the findings of the paper I think stem from the confusion between amplification host and reservoir host. I looked up a few other papers to try to get a sense for how this term is used elsewhere. At least two references I found (Weaver 2005, <https://www.ncbi.nlm.nih.gov/pubmed/16358422>; Vorou 2016 <http://www.sciencedirect.com/science/article/pii/S1201971216310578>) clarified the definition, but both of these differed (I think) from how Althouse et al. seem to be using the term. Does an amplifying host amplify viremia and therefore increase the likelihood that vectors become infected (the definition put forward by these two references)? Or does amplification refer to high levels of virus neutralization, such that amplification hosts both indicate that there is plenty of virus around that needs neutralizing, but in mounting this response they prevent onward transmission to vectors? It is clear that the authors are purporting the latter in their paper (abstract). But how is it an amplification host if it's not actually passing on the virus?

First let us say that the reviewer's confusion reflects a profusion of disparate definitions of reservoir and amplifying host in the field. This has been discussed explicitly in a 2017 review by Kuno et al. entitled *Vertebrate Reservoirs of Arboviruses- Myth, Synonym of Amplifier or Reality?* In this review, the authors state "It is stressed that in any branch of science the application of multiple definitions of a key term is a serious impediment to scientific communication, leading to unnecessary confusion".

We have inherited this problem of multiple definitions, and have sought to offer what Kuno et al. (2017) term a "pragmatic definition compatible with reality". Our definition of amplifying hosts hews closely to that of Weaver and Vorou. We write: "Alternatively, an arbovirus may initially be transmitted from the reservoir host into a different *amplification host* species, one that supports robust replication of the virus but is not necessary for persistence of the virus, and then from the amplification host to humans." This means, as the reviewer points out, the amplifying host amplifies the virus to high levels of viremia and increases the likelihood of

transmission to a mosquito and to other hosts (humans). We thank the reviewer for pointing out the references and have included them in this sentence.

In line 10, authors clearly define amplification hosts as those that support robust viral replication, and state that arbovirus transmission into humans is facilitated by these amplification hosts (this definition squares with the other two references). But in the abstract (third sentence from the end), authors state that because immunity (seropositivity) is too high, they must be amplification hosts because they do not support transmission. How can they be amplification host for humans if they are all immune, and therefore not passing CHIKV on to vectors? These statements seem in direct opposition to one another. If anything, the primates sampled in this study are very sensitive indicators that CHIKV is present in the environment – indeed, they seem more sensitive an indicator than primatophilic mosquitoes. But the NHPs are not playing a significant role in the transmission cycle of CHIKV in this system. Again, in line 68, authors state that high CHIKV seropositivity between amplifications in mosquitoes suggests that “host species other than monkeys serve as reservoirs in this area, while monkeys instead act as amplification hosts” – I agree with the first part – this evidence certainly suggests these species are not the reservoirs for CHIKV in this system. But the assertion that they must instead be amplifying virus seems counter to the previous assertion that these species are all immune and therefore not transmitting CHIKV to either vectors or to humans. This confusion followed me throughout the paper, and I had a hard time squaring lines 215-232 – if the majority of the populations are becoming infected and maintaining immunity starting at an early age (within the first year of life), how are these species at risk of initiating an explosive amplification of CHIKV?

We see how this is confusing. We consider monkeys in this region amplification hosts in that they cannot sustain ongoing CHIKV replication (high immunity after 1 year of life) but instead become transiently infected while young from the reservoir host population (as yet unidentified) which can then be transmitted to other host populations (humans, for example). Moreover high incidence of infection does seem to correlate temporally with amplifications in primatophilic mosquitoes; however low levels of infection still occur in juvenile monkeys even during inter-amplification periods.

We have modified the line in the abstract by inserting the word “alone” as below:

“... population seropositivity, and therefore immunity, was too high for monkeys alone to support continuous CHIKV transmission. We therefore conclude that monkeys in this region serve primarily as amplification rather than reservoir hosts of CHIKV”

I thought the field work and modeling were quite rigorous and clearly presented. I suspect that my lingering confusion over the inference from otherwise very straightforward empirical results could be easily remedied with clarifications in key places in the text. Perhaps a new definition of “amplification host” could specify how the

immune response (neutralizing persistent CHIKV, irrespective of amplifications detectable in mosquitoes) can be decoupled from transmission probability (the likelihood of serving as competent hosts from mosquito vectors). It seems to me like these two components of the definition are difficult to consider independently of one another because doing so would then lead to very different ecological outcomes.

We hope the explanation above helps the reviewer. We have modified the paragraph (lines 215-232) to clarify our meaning (ie, that monkeys born susceptible become infected rapidly which can then infect other species).

REVIEWERS' COMMENTS:

Reviewer #1 (Remarks to the Author):

Thank you for addressing my comments. Figure 2 should be ZIKV not ZIKAV. ONNV is not capitalized - not a proper name; top of page 10.

Reviewer #2 (Remarks to the Author):

The authors have addressed concerns sufficiently.

Reviewer #3 (Remarks to the Author):

The clarifications made in distinguishing amplifying hosts from reservoir hosts was very helpful and made the main arguments of this manuscript easier to follow.

REVIEWERS' COMMENTS:

Reviewer #1 (Remarks to the Author):

Thank you for addressing my comments. Figure 2 should be ZIKV not ZIKAV. ONNV is not capitalized - not a proper name; top of page 10.

We have fixed ZIKV and ONNV.

Reviewer #2 (Remarks to the Author):

The authors have addressed concerns sufficiently.

Reviewer #3 (Remarks to the Author):

The clarifications made in distinguishing amplifying hosts from reservoir hosts was very helpful and made the main arguments of this manuscript easier to follow.